# The Effects of Four Compounds That Act on the Dopaminergic and Serotonergic Systems on Working Memory in Animal Studies; A Literature Review

**DOI:** 10.3390/brainsci13040546

**Published:** 2023-03-25

**Authors:** Ștefania-Alexandra Grosu, Marinela Chirilă, Florina Rad, Andreea Enache, Claudia-Mariana Handra, Isabel Ghiță

**Affiliations:** 1Department of Child and Adolescent Psychiatry, “Prof. Dr Al. Obregia” Clinical Psychiatry Hospital, 041914 Bucharest, Romania; 2Pharmacy Faculty, “Titu Maiorescu” University, Bd. Gh. Şincai no.16, 040441 Bucharest, Romania; 3Department of Psychiatry, “Prof. Dr Al. Obregia” Clinical Psychiatry Hospital, 041914 Bucharest, Romania; 4Department of Occupational Medicine, Faculty of Medicine, “Carol Davila” University of Medicine and Pharmacy, Eroilor Sanitari Street 8, 050474 Bucharest, Romania; 5Department of Pharmacology and Pharmacotherapy, Faculty of Medicine, “Carol Davila” University of Medicine and Pharmacy, 050474 Bucharest, Romania

**Keywords:** working memory, bromocriptine, haloperidol, fluoxetine, ketanserin

## Abstract

The dopaminergic and serotonergic systems are two of the most important neuronal pathways in the human brain. Almost all psychotropic medications impact at least one neurotransmitter system. As a result, investigating how they affect memory could yield valuable insights into potential therapeutic applications or unanticipated side effects. The aim of this literature review was to collect literature data from animal studies regarding the effects on memory of four drugs known to act on the serotonergic and dopaminergic systems. The studies included in this review were identified in the PubMed database using selection criteria from the PRISMA protocol. We analyzed 29 articles investigating one of four different dopaminergic or serotonergic compounds. Studies conducted on bromocriptine have shown that stimulating D2 receptors may enhance working memory in rodents, whereas inhibiting these receptors could have the opposite effect, reducing working memory performance. The effects of serotonin on working memory are not clearly established as studies on fluoxetine and ketanserin have yielded conflicting results. Further studies with better-designed methodologies are necessary to explore the impact of compounds that affect both the dopaminergic and serotonergic systems on working memory.

## 1. Introduction

Memory is defined as the ability to retain and recollect information regarding previous experiences [1]. Following the reception of new data, there are three main actions: storage, consolidation, and retrieval. These actions are interconnected; however, retaining certain information does not necessarily guarantee the ability to access it later. This phenomenon can be explained by the vulnerability of recently acquired memories [1].

Working memory is a function that focuses on information which can be stored and used for performing cognitive tasks. The concept is usually presented in opposition with long-term memory, and thus, generally refers to short-term maintenance of information, with the possibility of it being updated flexibly [2]. Puig et al. (2014) [3] proposed different definitions of working memory depending on the species involved. In primates, working memory processes only last for a few seconds; in rodents, the term refers to a larger duration range, from a few seconds to hours. Thus, in rodents, working memory involves multiple mechanisms that could be classified as learning mechanisms in primates. 

The dopaminergic and serotonergic systems are two of the most important neuronal pathways in the human brain and they influence a wide variety of cerebral functions, including the modulation of memory and learning. Most psychotropic drugs affect at least one of the neurotransmitters from these systems, therefore studying their influence on memory could provide useful information regarding additional therapeutic benefits, or, on the contrary, raise awareness for potentially undocumented side effects.

Dopamine is a catecholaminergic neurotransmitter synthesized in three specific brain regions: the ventral tegmental area, the substantia nigra in the midbrain, and the arcuate nucleus in the hypothalamus. The main dopaminergic pathways of the brain are: the nigrostriatal pathway, part of the extrapyramidal system and involved in the motor function; the mesolimbic pathway, forming the reward system of the brain; the mesocortical system, hypothesized to be the origin of both cognitive and affective symptoms of schizophrenia; and the tuberoinfundibular pathway, involved in prolactin secretion [4]. 

Multiple studies have investigated the role of dopaminergic neurotransmission in memory processing, especially in patients with Parkinson’s Disease, most of them concluding that dopamine depletion impairs such functions [5,6]. In tasks which involve working memory, dopamine has a modulatory role on the neurons extending from the hippocampus to the prefrontal cortex through the D2 receptor [7]. This effect has been confirmed by administering D2 receptor agonists that improve, or antagonists, that impair working memory [8,9]. The fact that such compounds, which act on the D2 receptors, do not have any effects on working memory following intracortical injection, demonstrates that such receptors are not present in the prefrontal cortex. While the hippocampal dopaminergic system mediates the acquisition of new information, the cortical D1 receptors modulate the process of long-term potentiation, thus being involved in long-term memory [10].

Dopamine is involved in the processes of motivation, attention, and the positive reaction to reward. The former process has its neural correlation in the prefrontal cortex, while attention and positive reinforcement can be correlated with the nucleus accumbens. This fact can explain why motivation and reward make the learning process more effective [11]. 

Functional magnetic resonance imaging (fMRI) studies have correlated tasks involving working memory with the activation of the dorsolateral prefrontal cortex (DLPFC), although multiple brain areas need to be activated or inhibited to different degrees when the subject is performing such tasks [12]. Dopamine is a neurotransmitter with a crucial role in the prefrontal cortex, and studies have demonstrated that working memory performance depends on dopamine levels following an inverted U-shape, such that the highest performance is correlated with intermediate levels [13].

In human subjects, multiple studies have associated working memory deficits with psychiatric disorders such as schizophrenia, attention deficit hyperactivity disorder (ADHD) [14], bipolar affective disorder [15], and major depressive disorder [16]. Some authors have even identified such deficits to be predictive of the later onset of schizophrenia [17,18]. Administering dopaminergic or serotonergic compounds, that would normalize the amount of neurotransmitters and their respective receptors in specific brain regions could have an additional benefit in regulating such impairments, besides controlling the disease symptomatology.

Serotonin is a monoamine neurotransmitter synthesized in the raphe nuclei, located in the midbrain and pons. While there are only five types of dopamine receptors, serotonin receptors are much more diverse, seven classes of 5-HT receptors being known to date. The seven classes of serotonergic receptors can be divided into multiple subtypes. The most relevant in the process of memory formation is the 5-HT1A receptor, which acts as a modulator of other neurotransmitters, such as acetylcholine, gamma-aminobutyric acid (GABA), or glutamate. Such connections take place in the raphe nuclei, amygdala, septum, hippocampus, and cerebral cortex. Experiments on such receptors have demonstrated a pro-cognitive effect of their blockage, specifically using inverse agonists, thus facilitating glutamatergic neurotransmission, and on the other hand, their stimulation in human subjects has negative effects on explicit verbal memory [19]. Other serotonergic receptors, such as 5-HT1B, D and the class of 5-HT2, are involved in the acquisition and consolidation phases [10,11].

While serotonin has been less investigated than dopamine in the context of executive function regulation, multiple studies have investigated its role in modulating dopaminergic neurotransmission, specifically in the prefrontal cortex. This interaction involves the D2 and 5-HT2A receptors, and one review by Fink and Gothert (2007) indicated that blocking the 5-HT2A receptors increases dopamine release in the prefrontal cortex, indirectly [20].

Apart from modulation via certain neurotransmitters, there are other processes which can affect memory, such as neural degeneration. Aluminum, a neurotoxic substance, known to cause memory impairment, can cause both inhibition of various neurotransmitters (choline, GABA, glutamate, noradrenaline and serotonin) and neural degeneration (by decreasing the number of microchannels in the neurons, dendritic cells and various other cells involved in memory) [21]. 

The purpose of this review was to collect and evaluate literature data from studies done on animals regarding the effects of drugs known to act on the serotonergic and dopaminergic systems on memory.

## 2. Materials and Methods

The aim of this literature review is to investigate the effects on memory of 4 pharmaceutical compounds known to act on the serotonergic or dopaminergic systems: one agonist and one antagonist for each system considered.

### 2.1. Drugs Studied

The selection of the 4 pharmaceutical compounds to be included in this literature review was made considering the availability and amount of published data regarding their effects on working memory in animals. To identify which compounds were studied in this respect, a database search was conducted using the keywords: “dopamine agonists memory”; “dopamine antagonists memory”; “serotonin agonists memory”; “serotonin antagonists memory”. 

### 2.2. Literature Search Protocol

The literature search protocol was adapted from the PRISMA protocol recommendations. The eligibility criteria of the articles for this systematic review were selected based on the PRISMA protocol [22]. The studies were collected from the PubMed database in November 2022, and were selected using the following keywords: “bromocriptine memory”, “haloperidol memory”, “fluoxetine memory”, and “ketanserin memory”. Selection criteria included: open access to the article, animal studies not including any human subjects, both sexes of the species included in the study, and the language of the study being English, or any other language with an English translation available. Due to the scarcity of articles meeting all criteria, a filter for the year of publication was not added. 

### 2.3. Data Collection

Each study was manually evaluated by one reviewer. The data from each study that was analyzed in this review were the following: authors, year of publication, study design, animal model and species used, details about the pharmacological intervention: compound administered, dose and mode of administration, assessments used for evaluating working memory, and the results of the intervention on the animals’ performance in the working memory tests used.

## 3. Results

The compounds that had the most studies investigating their effects on working memory were bromocriptine and haloperidol, fluoxetine, and ketanserin.

Bromocriptine is a dopamine receptor agonist that has been studied extensively for its effects on working memory in animal models. Haloperidol is a dopamine receptor antagonist. Fluoxetine is a selective serotonin reuptake inhibitor. Ketanserin is a serotonin receptor antagonist.

The literature search conducted according to the predefined protocol identified 1078 articles in the PubMed database. One record was excluded because it was a duplicate. 970 records were excluded for not respecting eligibility criteria (any other study design such as review articles or studies without open access). Another 78 articles were excluded for the following reasons: irrelevant once full text was obtained (37), no existing English translation (2), human subjects (32), and studies investigating other types of memory, such as fear memory (7). After checking for duplicates and evaluating the studies according to the inclusion criteria, a total of 29 articles were selected and included in the final analysis. The Flow diagram is presented in Figure 1.

Out of the 29 articles included in this review, there were 4 articles related to bromocriptine, 10 articles related to haloperidol, 12 articles related to fluoxetine and 3 articles related to ketanserin. A summary tabulation of the articles selected and included in the analysis is presented in Table 1. 

### 3.1. Methods Used in the Studies 

Most of the studies relied on various types of mazes for the assessment of working memory. These tests are based on the animal’s capacity to retain spatial information based on specific cues. While they are easy to implement and measure, maze tests have the disadvantage that they require the animals to be stressed, which could affect the results of any kind of cognitive tasks [52].

The Morris water maze is based on the rodents’ aversion to water. It consists of a container filled with water, with an escape platform that is not directly visible. In order for the animal to escape, it must use certain visual cues, which are accessed using working memory. The parameter that the Morris water maze measures is the escape latency, which is the time necessary for the rodents to find the escape platform [53].

The Y and T mazes are based on the animals’ innate curiosity and interest in novelty. A spontaneous alternation is defined by the subject’s choice to explore novel areas, rather than those which the rodent had walked through before. Alternations can be encouraged by rewards, which are only accessed if the animal browses a new arm of the maze [54].

The radial arm maze consists of either 8 or 12 arms, which emerge radially from a central area, where the subject is initially placed. A reward is placed at the end of each of the arms, and the number of errors is counted, which are calculated as the number of times that the animal enters the same arm twice. Better working memory performance implies fewer errors [55]. 

The elevated plus maze is a cross-shaped maze, placed at a certain height above the ground. Two of its arms are open, and the other two are closed. This assessment is used to measure anxiety-like behavior, and therefore it is useful in testing the effects of anxiolitic compounds. Anxious subjects tend to prefer to stay in the closed arms, and have a lower tendency of walking through the open ones [56]. 

The Barnes maze is a round surface, with a number of holes close to its circumference. Under one of them there is an escape tunnel. The rodent needs several training sessions in the acquisition phase. When the subject is tested, the escape tunnel is removed, and multiple parameters are measured, such as the distance traveled by the animal before reaching the previous escape tunnel location [57].

The Wisconsin General Test Apparatus is a device used to study learning in primates. It consists of a tray, on which various objects are placed. Underneath one of the objects there is food reward. Between the subject and the investigator, there is a one way screen, allowing the researcher to observe the animal’s behavior [58].

### 3.2. Bromocriptine

Three out of the four articles identified for bromocriptine concluded that bromocriptine led to improved working memory in different types of maze tests [23,24,25]. The fourth study analyzed the effects of bromocriptine on memory-depleted rhesus monkeys and the authors concluded that bromocriptine impaired working memory both individually and paired with morphine [26]. Tarantino et al. [23], compared the effects of a SKF 38393, a D1 receptor agonist, with the effects of bromocriptine on working memory and reference memory in 12 mice. While D1 agonism did not affect either of the functions, bromocriptine improved working memory, and had no effect on reference memory. Phelps et al. analyzed the effects of various psychotropic compounds, on working memory in 60 rats subjected to traumatic brain injury. Bromocriptine was administered to 10 subjects, which were compared to a sham group of four rats. The D2 agonist had beneficial effects on spatial learning and memory retention when compared to haloperidol or risperidone [24]. Onaolapo et al. [25] administered different concentrations of bromocriptine, in two groups of 20 mice each, over a period of 21 days, and compared their effects on spatial working memory. The best performance was observed in the group that received 2.5 mg/kg of bromocriptine, when compared to the control group and another group that received a dose of 5 mg/kg. Finally, in a study including four rhesus monkeys, bromocriptine significantly decreased spatial working memory, the animals’ performance being measured before and after treatment [26].

### 3.3. Haloperidol

Five of the ten eligible studies included in this review used mice as test subjects, four studies used rats, and one article investigated the effects of haloperidol on working memory in monkeys. Two of the studies on rats, including a total of 204 subjects, revealed detrimental effects of haloperidol on working memory, in contrast to risperidone which was evaluated using different maze tests [32,33]. Another study concluded no statistically significant changes in performance after administering haloperidol to rats (8–10 subjects in each of the treatment and control groups), in either of the 1 mg/kg or 2 mg/kg doses [31]. Two other studies [27,28], one including 20 mice, and the other including 100 mice, used MK-801, an NMDA antagonist, to impair working memory in mice and examined whether haloperidol or olanzapine, an atypical antipsychotic, could reverse the damage. Haloperidol had no effects on the working memory tests applied in either study, unlike olanzapine, which yielded conflicting results. The study conducted by Ning et al. [28] also analyzed the effects of ziprasidone, a second generation antipsychotic, and PHA-543613, a nicotinic receptor agonist. Both compounds caused improvements in working memory performance. Boerner et al. [29] used 64 GluA1 knockout mice, lacking a fragment of the AMPA glutamate receptor, and administered LY354740, a glutamate receptor agonist and haloperidol. Consistent with the previous articles, haloperidol did not affect working memory test results in a genetic model of glutamatergic hypofunction, and neither did stimulating the glutamatergic receptors. Another study using 30 mutant mice, with reduced dopamine transporter (DAT) expression [30], yielded an interesting conclusion: a low dose of haloperidol was equivalent to the hyperdopaminergic state caused by the DAT mutation, in terms of working memory performance. Another article investigated the effects of intermittent versus continuous haloperidol or quetiapine administration in a total number of 70 rats with previously induced traumatic brain injury. The study concluded that continuous, but not intermittent administration of haloperidol affected the acquisition phase of spatial learning, while quetiapine did not influence it significantly [34]. One study used haloperidol specifically to impair memory in mice (80 total subjects), and this effect was proved using the novel object-recognition test, where haloperidol-treated mice had significantly poorer results when compared to the control group. This effect could be reduced by administering the histone deacetylase inhibitor tacedinaline (CI-994) [35]. The only study included in this review that assessed the effects of haloperidol on monkeys evaluated three macaca mulatta monkeys, and its results were consistent with most of the previous articles: learning rates were impaired by systemic injections of haloperidol [36].

### 3.4. Fluoxetine

Ten of the twelve studies included in this review investigated the effects of fluoxetine on previously induced memory loss through multiple interventions, such as: radiotherapy and temozolomide, a type of chemotherapy, [38]; intrahippocampal colchicine infusions [39]; bulbectomy [41]; using knockout mice models for schizoaffective disorder [42] and Alzheimer’s disease [46,48]; inducing different types of psychological stress [44,47]; and traumatic brain injury [45]; the other two included healthy subjects. A study by Flores-Ramirez et al. (2019), conducted on 103 subjects, found that fluoxetine pretreatment for 15 consecutive days, and assessing memory task performance an additional 3 weeks later, affected spatial working memory in males, but had no effect on performance in female mice [37]. Gan et al. (2019) [38] investigated the effects of fluoxetine on 36 mice previously trated with temozolomide, a chemotherapic drug, and radiotherapy. The mice in the experimental group showed increased anxiety-like behavior and cognitive impairment, which were reduced by chronic fluoxetine administration. Keith et al. (2007) [39] concluded that chronic fluoxetine treatment had no effect on working-memory task performance in a group of 39 rats who had previously received intrahippocampal colchicine injections, causing the destruction of a large portion of the granule cells in the dentate gyrus. Van Dyke et al. (2019) [40] suggested a detrimental effect of chronic fluoxetine administration on long-term memory recall in six healthy rats, compared to a control group of seven subjects. In a study by Zhou et al. (2019) including eight subjects, using a depression model consisting of olfactory bulbectomized rats, a fluoxetine trial could reverse the animals’ hyperemotional response to stimuli; however it could not reverse memory working memory impairments [41]. Fournet et al. (2012) used a knockout mouse model (60 subjects) for schizoaffective symptoms. Chronic fluoxetine administration had no effect on the number of spontaneous alterations in the Y-maze test; however, it improved short term memory [42]. The results of the study of Han et al. (2015), which included 40 rats, assessed the effects of a single dose of fluoxetine, previous and subsequent to the administration of a psychological stressor to the experimental group. The results suggested that fluoxetine could only improve working memory performance if it was administered preventatively, and not following the stress induction [43]. In 2017, Jayakumar et al. [44] investigated the effects of fluoxetine pretreatment on a total number of 18 rats that were consequently subjected to cold restrained stress. The pretreated group had superior working memory performance compared to the control group. Wang et al. (2011) compared 78 mice with traumatic brain injury, with a sham group of 70 mice. Both groups received a 4-week treatment with fluoxetine, and the antidepressant could not reverse the memory deficits produced; however, post-treatment microscopic modification was identified [45]. In a study by Ma et al. (2019), including 40 subjects, a transgenic model of Alzheimer’s disease mice received a 5-week treatment with fluoxetine, which alleviated spatial learning impairment and increased the number of neurons in the dentate gyrus [46]. Another study that used a transgenic mouse model for Alzheimer’s disease (60 total subjects) concluded that fluoxetine delayed declines in working memory and produced microscopic changes in the hippocampus [48]. Ibi et al. (2008) attempted to use fluoxetine in a social isolation mouse model (39 total subjects), and found that the treatment prevented impairment both in working memory tests, and in neuronal survival [47].

### 3.5. Ketanserin

Of the three articles on ketanserin included in this study, Levin et al. (2007) showed that doses of 0.5 mg/kg, 1 mg/kg, or 2 mg/kg caused a negative effect of the compound on working memory improvements produced by a 0.2 mg/kg dose of nicotine in a total number of 11 rats. However, the same doses of ketanserin did not affect memory performance when the animals received 0.4 mg/kg of nicotine [49]. A study by Aldridge et al. (2005) investigating 36 rats assessed the effects of ketanserin on working memory in rats treated with chlorpyrifos, an organophosphate. While it had no effects on the control group, ketanserin produced a dose-dependent increase in errors in the experimental group [50]. Finally, DeNoble et al. (1990) used eight squirrel monkeys, previously exposed to hypoxia, to evaluate the effects of ketanserin on working memory. Ketanserin, together with pirenperone and mianserin (other 5-HT2 antagonists drugs), produced significant improvements in working memory tests, while cyproheptadine, a nonselective serotonin antagonist, did not cause such effects [51].

## 4. Discussion

Bromocriptine is a dopamine receptor antagonist that has been studied extensively for its effects on working memory in animal models. Bromocriptine has the highest affinity for D2 receptors and acts primarily on the tuberoinfundibular pathway. Antagonists work by binding to receptors and inhibiting the effects of the same receptors’ agonists. Generally, their intracellular effect is to block messenger pathways [59]. Bromocriptine’s main clinical uses are in endocrinology, being the primary treatment for hyperprolactinaemia or acromegaly caused by pituitary adenomas [60]. Other less popular uses for bromocriptine are Parkinson’s disease [61] and type 2 diabetes [62]. The studies analyzed in this review showed that bromocriptine had beneficial effects on memory test performance in rodents, thus reflecting the specific importance of D2 receptors in spatial memory, unlike D1 receptors. The only study on bromocriptine which included monkeys as test subjects showed detrimental effects of the compound on working memory, which could either be explained by a species-specific effect or by the different concepts of the term ‘working memory’, depending on the species investigated.

Haloperidol is a dopamine receptor antagonist, acting particularly on the D2 receptors in the mesolimbic pathway; however, its effects on the tuberoinfundibular, mesocortical, and nigrostriatal pathways are responsible for its prevalent side effects [4]. It is mainly used as an antipsychotic in psychiatric disorders such as acute psychosis, schizophrenia, tic disorders, and specifically in children, for behavior problems or hyperactivity [63,64]. This review showed that haloperidol’s D2 receptor antagonism produced a decrease in the performance of animals in spatial memory tests. In this case, the result was homogenous throughout the different animal species included in the study. These results might suggest that dopamine in optimal quantities is necessary to maintain high performance in terms of working spatial memory.

Fluoxetine is the first selective serotonin reuptake inhibitor (SSRI) that was discovered, and is used as an antidepressant [65]. Fluoxetine works by inhibiting the serotonin transporter that aids in redirecting the neurotransmitter from the synaptic cleft back to the presynaptic membrane, thus increasing the intrasynaptic serotonin concentration [65]. More recent studies have revealed an effect on neuroplasticity of SSRIs by regulating the transcription of certain nuclear neurotrophic factors [66]. The main clinical uses for SSRIs, including fluoxetine, are for major depressive disorder, as well as other types of depressive disorders, obsessive-compulsive disorder, and off-label for generalized anxiety disorder, phobias, posttraumatic stress disorder and bulimia nervosa [64]. Analysis of the studies included in this review showed that the experiments using fluoxetine yielded contradicting results; some studies concluded beneficial results of the use of fluoxetine in brain damage models, while a similar number of studies found no effects of the compound. Increasing the amount of serotonin in the synaptic cleft did not improve mnesic function in animals without associated pathologies, but only in those with organic brain changes. This may also be relevant in the human species, in the case of patients treated with antidepressants from the SSRI class, who may notice improvements in mnesic function following chronic treatment.

Ketanserin is a serotonin antagonist, acting selectively on the 5-HT2 receptor. It is prescribed sparsely today, however, it is still used as an antiplatelet drug in patients over 55 [67], and as a vasodilator in both peripheral vascular disease and arterial or pulmonary hypertension. Another beneficial effect of ketanserin is the decrease of LDL cholesterol [68]. This review showed that ketanserin did not diminish the effects of a previous learning period, as expected, being beneficial even in the case of microscopic or macroscopic brain lesions. It should also be noted that the effects of ketanserin differed depending on the sex of the animals; the substance “normalizes” performance differences within the labyrinths, which are sex dependent. In the case of ketanserin, while the results were different within the two studies investigating rodents, the small number of articles included in this review makes it difficult to draw any conclusions regarding its effects on working memory, and whether they are species specific.

The results of this review are in line with the previous literature published, showing that both the dopaminergic and serotonergic systems play a role in learning and memory [69,70]. Dopamine is linked to learning and memory, as the stimulation of dopamine receptors is required for memories to be encoded and consolidated [71,72]. M.C. Buhot et al. (2000) also presented the effects of the administration of various serotonin receptor agonists and antagonists on memory [70].

Although one might hypothesize that a higher concentration of a neurotransmitter in the synaptic cleft would result in better working memory performance, the results of studies reviewed here do not consistently support this hypothesis, especially in the case of serotonin, where excess levels can impair cognitive function and memory. Further studies are needed to investigate the effects of different doses of serotonergic and dopaminergic compounds on working memory performance.

Both neurotransmitters appear to have an optimal concentration in the synaptic cleft for maximal working memory task performance. Therefore, the results of this study regarding dopamine are consistent with the study conducted by Seamans and Yang [13], where working memory performance is dependent on dopamine levels in an inverted U-shaped curve manner. In the case of serotonin, this review confirmed the results of multiple previous studies that demonstrated its neurotoxicity when it is secreted in excess. It should also be noted that different serotonergic receptors have different functions in terms of their excitatory or inhibitory nature, respectively, and this review considered the use of non-selective substances to measure the effects on all receptors of the neurotransmitter.

Therefore, the results of this review suggest that an optimal concentration of either neurotransmitter is needed for maximal working memory task performance in animal studies. Moreover, the most significant instance when drugs could improve working memory is if associated organic disruptions in the subject’s brain are already present. Such disruptions could potentially benefit from pharmacologic intervention, targeting pathways such as the dopaminergic or serotonergic systems.

A practical example of drugs that act on both serotonergic and dopaminergic receptors and have effects on the functions correlated with the prefrontal cortex are atypical antipsychotics. These compounds bind to multiple serotonergic and dopaminergic receptors, thus controlling psychotic symptoms in patients suffering from disorders such as schizophrenia. Other than the known positive symptoms of psychotic disorders, cognitive disruptions, including working memory deficits, can also be identified in such patients [73].

Having a variable affinity for 5-HT2A receptors, but with a homogenous antagonistic effect, atypical antipsychotics may have an additional benefit of improving cognitive symptoms, including working memory impairment, by specifically increasing dopamine secretion in the area of the brain that is predominantly correlated with cognition, namely the prefrontal cortex [4].

Taking into consideration the different paradigms of working memory, depending on the species investigated, as mentioned in the introduction, it is important to note that studies of working memory should be compared only within the same class of species, and a common, clearly defined terminology should be adopted to avoid cross-species confusion.

It is also important to note that the effects of a drug on working memory may vary depending on the specific task and the dosage used.

Animal research plays an important role in investigating the effects of drugs on cognitive processes such as working memory. It can provide a framework to gain valuable insights into the underlying mechanisms of these processes and how they may be affected by various drugs. It can also help identify potential safety concerns or adverse effects of drugs on cognitive processes and determine the therapeutic benefits of drugs for cognitive impairment.

Therefore, animal studies can be used for the identification of certain brain circuits involved in processes such as working memory, as well as the influence of different drugs, both on the respective circuits and on the subject’s behavior. Such studies are easier to conduct than those involving human subjects, and having the possibility of obtaining animal models for specific psychiatric disorders, animal studies are crucial in the first steps of developing new drugs. Apart from identifying underlying mechanisms or physiological and pathological processes, animal subjects also offer insight into potential adverse effects and safety profiles of newly developing compounds.

However, animal research also has limitations. For instance, findings from animal studies may not always translate directly to humans due to differences in physiology and biology. Therefore, animal research should be used in combination with other approaches, such as human clinical trials and in vitro studies, to gain a comprehensive understanding of the effects of drugs on cognitive processes. Additionally, ethical considerations should be considered to ensure the humane treatment of animals used in research.

The use of animal models in behavioral research has been criticized, and even though such models usually have good face validity, it is difficult to assess them in terms of predictive or construct validity [74,75]. Future research can integrate the findings of animal studies into more complex human studies, possibly with the aid of newer tools such as functional neuroimaging. Recent research advances have facilitated the use of brain organoids for more in-depth neurobiological studies.

An important limitation of this study is that only one type of memory could be investigated, due to a lack of standardization for the evaluation of other types in non-human species. Therefore, multiple meta-analyses and systematic reviews should be performed, investigating studies including exclusively human subjects, and evaluating multiple types of memory.

Identifying a specific psychological pattern induced by pharmacologic manipulation of neurotransmitters could offer additional aid in alleviating symptoms associated with psychiatric disorders such as depression or schizophrenia. These types of studies could help in understanding the pathophysiology of such disorders, with the hope that one day enough information will be collected to facilitate an etiological treatment in favor of a symptomatic one.

No study risk of bias assessment or certainty assessment was conducted. This review included a small number of studies, which were heterogenous regarding both the species included and the assessments used to evaluate working memory. In addition, this review presented an interpretation of the outcome of each study included based on the reported statistically significant results. However, the statistical significance has the great limitation that it depends on sample sizes. Therefore, the conclusions for each compound studied may not be accurate, and a different study design, such as a systematic review or meta-analysis should be applied, highlighting the need for statistical analysis and further interpretation of the results.

## 5. Conclusions

This literature review provides an overview of studies examining the impact of four compounds that affect the dopaminergic and serotonergic systems on working memory in animals. The findings suggest that stimulating the D2 receptors with bromocriptine may enhance working memory in rodents, whereas blocking these receptors can reduce working memory performance. However, the effects of serotonin on working memory are inconclusive due to conflicting results from studies on fluoxetine and ketanserin. More research with improved methodologies is necessary to better understand the impact of compounds that affect both the dopaminergic and serotonergic systems on working memory.

## Figures and Tables

**Figure 1 brainsci-13-00546-f001:**
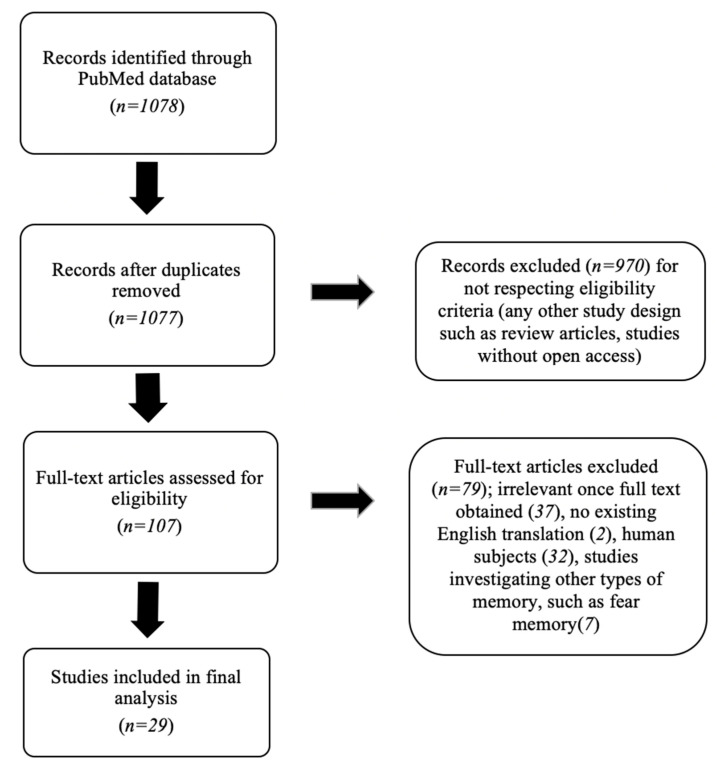
Flow diagram of the study.

**Table 1 brainsci-13-00546-t001:** Summary table of the articles selected and included in the review.

Author (Year)	Animal Species	Intervention	Assessments	Results
Tarantino et al. (2011) [23]	C57BL/6N Mice	Bromocriptine, 10 mg/kg i.p.	12-arm Radial maze	Improved working memory
Phelps et al. (2015) [24]	Sprague-Dawley rats	Bromocriptine, 5 mg/kg i.p.	Water maze	Improved working memory
Onaolapo et al. (2013) [25]	Swiss albino mice	Bromocriptine, 2.5, and 5 mg/kg p.o.	Y-maze	Improved working memory (only for doses of 2.5 mg/kg)
Wang et al. (2013) [26]	Rhesus monkeys	Bromocriptine, 1 mg/kg p.o.	Wisconsin General Test Apparatus	Impaired working memory
Song et al. (2016) [27]	C57BL/6J mice	Haloperidol, 0.05 mg/kg; MK-801, 0.1 mg/kg i.p.;	Morris water maze	No effect
Ning et al. (2017) [28]	C57BL/6J mice	Haloperidol, 2 mg/kg; MK-801, 0.1 mg/kg i.p.	Morris water maze	No effect
Boerner et al. (2017) [29]	Gria1^−/−^ mice	Haloperidol, 0.3 mg/kg i.p.	T-maze	Impaired working memory
Deng et al. (2015) [30]	DAT^+/−^ mice	Haloperidol, 0.02 mg/kg i.p.	T-maze	Impaired working memory
Hutchings et al. (2013) [31]	Wistar rats	Haloperidol, 1 and 2 mg/kg p.o.	Radial arm maze	No effect
Terry et al. (2007) [32]	Wistar rats	Haloperidol, 2 mg/kg p.o.	Water maze	Impaired working memory
Terry et al. (2007) [33]	Wistar rats	Haloperidol, 2 mg/kg p.o.	Water maze; Radial arm maze	Impaired working memory
Weeks et al. (2019) [34]	Rats	Haloperidol, 0.5 mg/kg i.p.	Morris water maze	Impaired working memory
McClarty et al. (2021) [35]	C57BL/6 mice	Haloperidol, 0.05 mg/kg i.p.	Novel object recognition test	Impaired working memory
Turchi et al. (2010) [36]	Macaca mulatta monkeys	Haloperidol, 10 and 17.8 μg/kg i.m.	Touching a picture on a screen to receive a reward	Impaired working memory
Flores-Ramirez et al. (2019) [37]	C57BL/6 mice	Fluoxetine, 20 mg/kg p.o.	Morris water maze	Impaired working memory in males; no effect in females
Gan et al. (2019) [38]	C57BL/6J mice	Fluoxetine, 5 mg/kg i.p.	Water maze; Cross maze	Improved working memory
Keith et al. (2007) [39]	C57BL/6J mice	Fluoxetine, 5 mg/kg i.p.	Morris water maze	No effect
Van Dyke et al. (2019) [40]	C57BL/6J mice	Fluoxetine, 80 mg/L in drinking water, with an average consumption of 1.2–1.6 mg of fluoxetine per day	Morris water maze	Improved working memory
Zhou et al. (2019) [41]	Sprague-Dawley rats	Fluoxetine, 10 mg/kg p.o.	Morris water maze	No effect
Fournet et al. (2012) [42]	Knockout mice for the STOP gene	Fluoxetine, 10 mg/kg p.o.	Y-maze	No effect
Han et al. (2015) [43]	Sprague-Dawley rats	Fluoxetine, 10 mg/mL p.o.	Morris water maze	Improved working memory, only when administered before the stressor
Jayakumar et al. (2017) [44]	Wistar rats	Fluoxetine, 10 mg/kg p.o.	Morris water maze	Improved working memory
Wang et al. (2011) [45]	C57BL/6 mice	Fluoxetine, 10 mg/kg i.p.	Barnes maze	No effect
Ma et al. (2019) [46]	APPswe/PSEN1dE9 mice	Fluoxetine, 10 mg/kg i.p.	Morris water maze; Y-maze	Improved working memory
Ibi et al. (2007) [47]	Mice	Fluoxetine, 10 mg/kg i.p.	Water maze	Improved working memory
Chao et al. (2021) [48]	APP/PS1 mice	Fluoxetine, 10 mg/kg i.p.	Open-field test, Morris water maze, Y-maze	Delayed declines in working memory
Levin et al. (2007) [49]	Rats	Ketanserin, 0.5, 1, and 2 mg/kg p.o.	Radial arm maze	Impaired working memory
Aldridge et al. (2005) [50]	Sprague-Dawley rats	Ketanserin, 0.5, 1 and 2 mg/kg p.o.	Radial arm maze; Cross maze	Improved working memory only in animals who had been previously administered chlorpyrifos
DeNoble et al. (1990) [51]	Squirrel monkeys	Ketanserin, 0.1 mL/0.1 kg p.o.	Pressing the correct button to receive a reward	Improved working memory, only in animals previously exposed to hypoxia

## Data Availability

The dataset used and/or analyzed during this study is available from the corresponding author upon reasonable request.

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
