# Peer review of "The Effects of Four Compounds That Act on the Dopaminergic and Serotonergic Systems on Working Memory in Animal Studies; A Literature Review"

_brainsci, 2023, doi:10.3390/brainsci13040546_

Round 1

Reviewer 1 Report

The study conducts a systematic review of the effect of dopaminergic and serotonergic systems on memory functioning in animal studies. I consider that it is not a systematic review but rather a classic narrative review where the literature search conforms to the standards of a systematic review under the PRISMA protocol. The definition of a systematic review can be found in the “Cochrane handbook for systematic reviews of interventions (Higgins et al. 2019)1:

A systematic review attempts to collate all empirical evidence that fits pre-specified eligibility criteria in order to answer a specific research question. It uses explicit, systematic methods that are selected with a view to minimizing bias, thus providing more reliable findings from which conclusions can be drawn and decisions made (Antman 1992, Oxman1993). The key characteristics of a systematic review are:

·         A clearly stated set of objectives with pre-defined eligibility criteria for studies;

·         an explicit, reproducible methodology.

·         A systematic search that attempts to identify all studies that would meet the eligibility criteria.

·         An assessment of the validity of the findings of the included studies, for example through the assessment of risk of bias.

·         A systematic presentation, and synthesis, of the characteristics and findings of the included studies.”

I consider the authors have identified the articles in line with what would be a systematic review but the process of extracting quantitative information from the studies, as well as the assessment of the possible biases that may occur have been omitted. The presentation of the quantitative information of the studies is based on the "vote-counting method" procedure used in narrative reviews and which has been much criticized (Hedges & Olkin, 2014)2.  The quality of the studies retrieved has not been discussed, neither was the publication bias. For these reasons I think this manuscript should not be accepted for publication in the journal.

1.-Higgins, J. P., Thomas, J., Chandler, J., Cumpston, M., Li, T., Page, M. J., & Welch, V. A. (Eds.). (2019). Cochrane handbook for systematic reviews of interventions. John Wiley & Sons.

2.-Hedges, L. V., & Olkin, I. (2014). Statistical methods for meta-analysis. Academic press.

Author Response

Response to Reviewer 1 Comments

“The study conducts a systematic review of the effect of dopaminergic and serotonergic systems on memory functioning in animal studies. I consider that it is not a systematic review but rather a classic narrative review where the literature search conforms to the standards of a systematic review under the PRISMA protocol. The definition of a systematic review can be found in the “Cochrane handbook for systematic reviews of interventions (Higgins et al. 2019).

I consider the authors have identified the articles in line with what would be a systematic review but the process of extracting quantitative information from the studies, as well as the assessment of the possible biases that may occur have been omitted. The presentation of the quantitative information of the studies is based on the "vote-counting method" procedure used in narrative reviews and which has been much criticized (Hedges & Olkin, 2014).  The quality of the studies retrieved has not been discussed, neither was the publication bias. For these reasons I think this manuscript should not be accepted for publication in the journal.”

Authors’ comment: the manuscript was amended following the above observations to mention literature search instead of systematic review and to provide more details on the studies included in the study. 

Reviewer 2 Report

References should be uniformly presented; some have all authors but other have "et al" abbreviations

Author Response

Response to Reviewer 2 Comments

References should be uniformly presented; some have all authors but other have "et al" abbreviations.

Author’s comment: the references were corrected according to the observation above.

Reviewer 3 Report

This manuscript conducted a systematic review to investigate the effects of four compounds known to affect the serotonergic and dopaminergic systems on working memory in animal studies. This study does present some interesting results but there are several issues that need to be addressed further.

Introduction:

·       It seems to be inappropriate to use two paragraphs on the discussion of nicotine in the introduction (from line 43 to line 57). There is a lack of a strong correlation between the topic of this study and the subject of nicotine. Two references cited here (reference 2 and 3) are from the co-authors.

Methods:

·       Please provide essential information/references for selecting the four compounds that are “commonly used in clinical practice” in the “drugs studied” session.

·       Regarding the “literature search protocol”, please

o   provide a detailed research strategy instead of “initial keywords”.

o specify the selection process as well as the data collection process, such as the number of reviewers and the fashion of how they worked for each process.

o   provide references on the statement of “how often the substance is used clinically” (line 101) and details on “the number of articles yielded for every case” (line 102).

o  the PRISMA protocol has been updated to the 2020 version, please revisit your protocol, and revise accordingly.

Results:

·       Please provide details on the exclusion reasons for the 970 records shown in Figure 1 and specified numbers for each reason for the 78 records as well.

·       As for Table 1, please

o   provide meaningful information according to the table header. For instance, the authors provided information on the animal species instead of the animal model, please refine the content accordingly. Besides, if all included studies are experimental studies, the third column seems quite unnecessary.

o   please follow the guidelines on formatting tables to revise table 1.

·       Please check the references on line 124. There should be three instead of one. Similar problems related to the inaccuracy of the citations also occur on lines 132, 137, and 154-157.

Discussion and conclusion:

·       Regarding the heterogeneity of included experimental studies on animal species and models as well as methods for drug delivery, it is hard to draw conclusions that a) this review could reject the theory that “higher concentration of a neurotransmitter is in the synaptic clef, the better the performance in working memory tests” (line 227-230), b) the substances that act on the serotonergic and dopaminergic systems have beneficial effects when used in “medium doses” (line 263-4). Besides, only a few included studies in this review were designed in a way that the effects of different concentrations of one certain drug on working memory could be investigated appropriately.

·       Please provide a reference for line 196.

Others:

·       The title is too broad, please refine the “dopaminergic and serotonergic systems”.

·       It is difficult to understand the lines from 238 to 242. Please revise this long sentence.

Author Response

Response to Reviewer 3 Comments

“This manuscript conducted a systematic review to investigate the effects of four compounds known to affect the serotonergic and dopaminergic systems on working memory in animal studies. This study does present some interesting results but there are several issues that need to be addressed further.

Introduction:

  • It seems to be inappropriate to use two paragraphs on the discussion of nicotine in the introduction (from line 43 to line 57). There is a lack of a strong correlation between the topic of this study and the subject of nicotine. Two references cited here (reference 2 and 3) are from the co-authors. “

Author’s comment: the paragraph was amended following the observation above.

“Methods:

  • Please provide essential information/references for selection of the four compounds that are “commonly used in clinical practice” in the “drugs studied” session.”

Author’s comment: the manuscript was amended to include the process of selection of the 4 compounds.

  • “Regarding the “literature search protocol”, please
    • provide a detailed research strategy instead of “initial keywords”.
    • specify the selection process as well as the data collection process, such as the number of reviewers and the fashion of how they worked for each process.
    • provide references on the statement of “how often the substance is used clinically” (line 101) and details on “the number of articles yielded for every case” (line 102).
    • the PRISMA protocol has been updated to the 2020 version, please revisit your protocol, and revise accordingly.”

Author’s comment: the manuscript was amended to include clarifications and a more detailed description of the literature search protocol based on the observations above.

“Results:

  • Please provide details on the exclusion reasons for the 970 records shown in Figure 1 and specified numbers for each reason for the 78 records as well.
  • As for Table 1, please
    • provide meaningful information according to the table header. For instance, the authors provided the information on the animal species instead of the animal model, please refine the content accordingly. Besides, if all included studies are experimental studies, the third column seems quite unnecessary.
    • please follow the guidelines on formatting tables to revise table 1.
  • Please check the references on line 124. There should be three instead of one. Similar problems related to the inaccuracy of the citations also occur on lines 132, 137, and 154-157.”

Author’s comment: the manuscript was corrected based on the observations above.

“Discussion and conclusion:

  • Regarding the heterogeneity of included experimental studies on animal species and models as well as methods for drug delivery, it is hard to draw conclusions that a) this review could reject the theory that “higher concentration of a neurotransmitter is in the synaptic clef, the better the performance in working memory tests” (line 227-230), b) the substances that act on the serotonergic and dopaminergic systems have beneficial effects when used in “medium doses” (line 263-4). Besides, only a few included studies in this review were designed in a way that the effects of different concentrations of one certain drug on working memory could be investigated appropriately.
  • Please provide a reference for line 196. “

Author’s comment: the manuscript was corrected based on the observations above.

“Others:

  • The title is too broad, please refine the “dopaminergic and serotonergic systems”.
  • It is difficult to understand the lines from 238 to 242. Please revise this long sentence.”

Author’s comment: the manuscript was amended based on the observations above.

Round 2

Reviewer 1 Report

The authors have removed the term “systematic review” and this better defines this sort of review. However, the changes made to the manuscript I believe are still insufficient. The limitation is that the authors continue to conclude from the significance of the results of the different articles reviewed. Statistical significance has the great limitation that it depends on sample sizes, so this process of extracting information from the articles has been highly criticized. This limitation should be included in the discussion and therefore the conclusions obtained should be less categorical. In my opinion the review is correct but I think that the value of the results is poor, a systematic review with meta-analysis should really be done. 

Author Response

Dear Reviewer 1,

We acknowledge and appreciate all the guidance provided. The limitations were included in the discussion and the conclusion was revised to be less categorical. This is a lesson learned that will improve our future work.

Kindly regards,

Authors

Reviewer 3 Report

None.

Author Response

Dear Reviewer 3,

We acknowledge and appreciate all the guidance provided.

Thank you for your support.

Authors